# Bacterial and Viral Infections in Liver Transplantation: New Insights from Clinical and Surgical Perspectives

**DOI:** 10.3390/biomedicines10071561

**Published:** 2022-06-30

**Authors:** Nour Shbaklo, Francesco Tandoi, Tommaso Lupia, Silvia Corcione, Renato Romagnoli, Francesco Giuseppe De Rosa

**Affiliations:** 1Department of Medical Sciences, Infectious Diseases, University of Turin, 10124 Turin, Italy; silvia.corcione@unito.it (S.C.); francescogiuseppe.derosa@unito.it (F.G.D.R.); 2General Surgery 2U and Liver Transplant Unit, Molinette Hospital, A.O.U. Città della Salute e della Scienza, University of Turin, 10124 Turin, Italy; francesco.tandoi@unito.it (F.T.); renato.romagnoli@unito.it (R.R.); 3Unit of Infectious Diseases, Cardinal Massaia Hospital, 14100 Asti, Italy; tommaso.lupia89@gmail.com; 4School of Medicine, Tufts University, Boston, MA 02111, USA

**Keywords:** liver transplant, bacterial infections, viral infections, infection control, screening, immunosuppression, surgical frontiers, liver perfusion

## Abstract

End-stage liver disease patients undergoing liver transplantation are prone to develop numerous infectious complications because of immunosuppression, surgical interventions, and malnutrition. Infections in transplant recipients account for the main cause of mortality and morbidity with rates of up to 80%. The challenges faced in the early post-transplant period tend to be linked to transplant procedures and nosocomial infections commonly in bloodstream, surgical, and intra-abdominal sites. Viral infections represent an additional complication of immunosuppression; they can be donor-derived, reactivated from a latent virus, nosocomial or community-acquired. Bacterial and viral infections in solid organ transplantation are managed by prophylaxis, multi-drug resistant screening, risk assessment, vaccination, infection control and antimicrobial stewardship. The aim of this review was to discuss the epidemiology of bacterial and viral infections in liver transplants, infection control issues, as well as surgical frontiers of ex situ liver perfusion.

## 1. Introduction

Liver transplantation is a cornerstone therapy for acute and chronic end-stage liver diseases [1]. End-stage liver disease patients assigned to transplantation are prone to develop numerous infectious complications because of immunosuppression, surgical procedures, and malnutrition [2,3]. Increased gastrointestinal permeability and pathological bacterial translocation along with prolonged hospitalizations and invasive devices place this population at risk for nosocomial infections such as pneumonia, central venous catheter (CVC)-related bacteraemia, and urinary tract infections [4,5].

Infections in transplant recipients account for the main cause of mortality and morbidity with rates of up to 80% [6,7,8], despite advanced surgical techniques, new immunosuppressive drugs, prophylactic antibiotics, vaccination, and infection control strategies [1]. Bacterial infections represent up to 70% of all infections in liver transplants, followed by fungal and viral infections [9]. The virulence of the pathogen, along with the intensity and timing of exposure, can also impact the severity and outcome of the infection. Factors identified to increase the risk of infections after liver transplantation include a Model for End-Stage Liver Disease (MELD) score above 30, reoperation, renal replacement therapy, prolonged intensive care unit (ICU) stay, older age, biliary tract manipulation, and invasive procedures [10]. However, for many opportunistic infections, effective prediction and prevention practices are lacking [11].

The aim of this review was to discuss the epidemiology of bacterial and viral infections in liver transplants, infection control issues, as well as surgical frontiers of ex situ liver perfusion.

## 2. Infection Control in Liver Transplant Recipients: Hygiene, Screening, Vaccination

The main difference between the impact of infection control among liver transplant patients and other hospitalized patients is the extent of benefit achieved by preventing infections linked to immunosuppression [12]. The basics of infection control in transplant patients begin with strategies that most centres follow in their inpatient and ambulatory care setting. Key efforts must concentrate on measures that are fundamental to infection prevention: infectious disease epidemiology, outbreak investigation, hand hygiene, screening programs, isolation practices, proper use of personal protective equipment (PPE), management of invasive devices and vaccination [13]. Procedures for sterilization and disinfection of the environment and equipment are well described in guidelines [14,15], but are often missed like other critically important aspects of infection control policies. Table 1 summarizes risk assessment and infection control procedures in solid organ transplant (SOT) during the stages of transplantation.

Additional efforts to protect patients from airborne fungi, respiratory viruses, gastrointestinal viruses and multi-drug resistant (MDR) Gram-negative bacteria are essential [13]. Guidelines have particularly addressed infection control of carbapenem-resistant organisms, recommending active surveillance of high-risk units and contacts with infected patients and contact isolation [16]. Cohorting these high-risk patients in single units can ensure consistency with transplant-specific infection prevention and that experienced staff monitor these patients [13].

Attention should be given to specific infection prevention policies such as: frequent air exchanges, high efficiency particulate air (HEPA) filtration in inpatient units, construction and renovation restrictions, environmental controls, avoidance of ornamental plants, filter and airway duct policies, staff and visitor screening, water management plans and *Legionella* screening [13].

In addition, patients should be assessed for risk of infection by a thorough medical history, including details of previous infections, places of travel and residence, lifestyle, and exposures to animals and environmental pathogens to evaluate the probability for reactivation of latent infection after transplantation. The pre-transplant period is the best time for comprehensive counselling of the patient and family about infection prevention practices including activities to avoid foodborne pathogens, pets, travel, post-exposure prophylaxis and immunization [17].

Infectious disease consultants, infection preventionists, and hospital epidemiologists are critical to protect transplant patients from major pathogens. Such teams should collaborate with primary transplant teams, hospital and nursing administration, and construction contractors to ensure the advance of prevention efforts [13].

### 2.1. Candidate Screening

Knowledge of pre-transplant colonization in donor and recipient helps to develop an individualized peri-transplant prophylactic antimicrobial regimen and optimize post-transplant outcomes. Risk evaluation based on detailed history and appropriate diagnostic assessment is necessary [17].

Guidelines for pre-transplant screening have been developed by several national and international multidisciplinary transplant groups [18,19]. The Centers for Disease Control and Prevention (CDC) has published guidelines for the prevention of HIV, HBV, and HCV transmission through organ transplantation [20]. In addition to serology, nucleic acid testing for hepatitis B, hepatitis C and HIV is required for deceased and living donors [17].

The Organ Procurement and Transplantation Network (OPTN) and United Network for Organ Sharing (UNOS) demands testing for CMV, EBV, toxoplasmosis and syphilis for deceased donors along with blood and urine testing. For HCV screening in dead donors, nucleic acid testing is employed with serological tests. CMV, EBV, TB, toxoplasmosis, and syphilis are also compulsory screenings for living donors. Several endemic exposures may require additional evaluation beyond recommended standard testing, such as *West Nile virus*, *Strongyloides*, *Coccidiodes*, and *Trypanosoma cruzi* [17,18,19,20,21].

Candidate screening also determines immunity to vaccine-preventable diseases and aids in assigning infected organs to recipients with proven immunity to certain organisms [18].

Several bacterial pathogens may be transmitted during transplantation [17]. Therefore, screening for MDR must be performed to achieve prompt isolation including screening for: carbapenem-resistant Enterobacteriaceae (CRE), *C. diff*, MRSA, VRE and other MDR Gram-negative rods (GNRs) [22]. Moreover, all candidates should have Mycobacterium tuberculosis testing with purified protein derivative (PPD) or interferon-gamma release assays (IGRAs) prior to transplant, and those who have a positive test or a history of active tuberculosis should undergo additional screening to exclude active disease [17] or need to be evaluated for prophylaxis.

### 2.2. Vaccination

Before transplantation, vaccination should be evaluated in detail, and any essential vaccinations should be administered. Vaccines should be given as early in the pre-transplant period as possible, when the probability of acquiring a protective immune response is the greatest and live vaccines can be given safely. After transplantation, inactivated vaccines are considered safe, although their efficiency may be reduced [23]. Although there is no agreement on the best time to vaccinate after transplantation, most centres resume vaccination at approximately 6 months after transplantation in patients who are on standard immunosuppressive therapy [24,25]. Many transplant centres perform routine pre-transplant serology for vaccine-preventable diseases such as hepatitis B, varicella, measles, mumps and rubella to direct individual vaccine recommendations [26,27].

All organ transplant patients should be vaccinated against HBV. Vaccination against seasonal influenza virus should be given yearly both pre- and post-transplantation. Transplant donors and recipients are recommended to be vaccinated against pneumococcus with both the pneumococcal conjugate vaccine and the pneumococcal polysaccharide vaccine. Vaccination against herpes zoster is required for transplant candidates who are older than 50 years [27].

Transplant candidates and recipients are eligible for anti-SARS-CoV-2 vaccine, and this is strongly recommended by scientific societies [28,29,30]. Additional vaccines may be required for patients depending on age, vaccination history, or other risk factors (e.g., meningococcal vaccination for patients given eculizumab) [31]. It is also necessary to determine if the donor has received live vaccination during the past 4 weeks against influenza, varicella, measles, mumps, rubella, Bacillus Calmette–Guérin (BCG), cholera, yellow fever and Salmonella typhi or polio [27].

## 3. Bacterial and Viral Infections in Liver Transplant Recipients

In liver transplant recipients, the risk of infection fluctuates over time [30]. The challenges faced in the early post-transplant period tend to be linked to transplant procedures and nosocomial infections [30,31]. Because of the increased burden of immunosuppression, opportunistic infections become more likely between 1–12 months post-transplant. As immunosuppression tends to decrease 12 months post-transplant, so does the risk of opportunistic infections [30,31,32,33].

Transplant recipients, however, remain at risk of contracting community-acquired infections, and recurrent cholangitis may become a problem in those with chronic allograft malfunction or recurrent cholestatic liver disease [4,34].

Antimicrobial resistance patterns are becoming more common among bacteria, but there are regional and centre-specific variances in prevalence rates [4,31]. Gram-positive infections were the leading cause of liver transplantation infectious complications; as many as 80% of patients are infected with the methicillin-resistant *S. aureus* (MRSA) bacteria, while 55% are infected with the vancomycin-resistant *Enterococcus* (VRE). Moreover, linezolid-resistant VRE outbreaks have been observed [4,31,34]. However, the increase in Gram-negative rods, mainly carbapenem-resistant *Enterobacteriales*, extended-spectrum beta−lactamase (ESBL), MDR *P. aeruginosa* and *Acinetobacter* has become a crucial concern in clinical practice [4,31,34].

### 3.1. Bacterial Infections

After liver transplantation, bacterial pathogens are the most common cause of infectious complications. Most of these infections occur in the first month following liver transplantation (Figure 1); they are more likely to affect the surgical site, abdomen, bloodstream, and urinary or respiratory tracts [4,34,35].

General risk factors for infectious complications after transplantation include biliary tract manipulation, the necessity for surgical and other invasive procedures, prolonged hospitalization, prior colonisation, mechanical ventilation, indwelling vascular and urinary catheterisation, and the microbiological status of the patient [4,34,35].

While infections in liver transplant recipients may be theoretically due to any bacterium, *Enterococcus* spp., *Streptococcus viridans*, *Staphylococcus aureus*, and other members of the *Enterobacteriaceae* family account for the vast majority of infectious diseases in this cohort [4,31,35].

From the last decade, we have witnessed a rise in MDR infections among cirrhotic and liver transplant patients. This is the result of the widespread use of antibiotic prophylaxis, frequent hospitalizations and higher rates of ICU admissions [4,31,35]. Salerno et al. [36] described in a multicentre study that extended-spectrum lactamase *E. Coli* and carbapenem-resistant *K. Pneumoniae* were the most common cause of infections in cirrhotic patients. Alexopoulou et al. [37] found 19% of MDR infections in the cirrhotic population with spontaneous bacterial peritonitis, which were linked to healthcare-acquired infections and higher MELD scores (28 vs. 19, *p* = 0.012). Moreover, Merli et al. [38] confirmed that MDR infections are more common in the hospital (56 percent in hospital-acquired/healthcare-associated infections vs. 22 percent in community-acquired infections, *p* = 0.008) in a multicentre prospective survey.

However, the majority of case series reporting rates of MDR Gram-negative infections in solid organ transplant patients were from endemic areas, resulting in quite high percentages ranging from 18% to 50% [4,39,40].

When considering MDR bacteria colonization, it is vital to note that bloodstream infections can spread in the post-operative period and/or when immunosuppression is introduced. Giannella et al. [41] assessed CR-KP colonization in 237 patients awaiting LT, of which 11 (4.6%) had positive rectal swabs at the time of LT. After LT, the CR-KP infections became active due to the following factors: hospitalization, higher MELD at LT, prior antibiotic exposure, postoperative difficulties, and time spent in the ICU.

### 3.2. Surgical Site Infections

Post-operative surgical site infections, which occur in approximately 10% of patients following liver transplantation, are one of the most prevalent bacterial infections [4,31,35]. Liver transplant recipients who require a significant number of blood transfusions are more likely to develop a surgical site infection, which suggests a more complex surgical process and a longer recovery period for the patient [35]. Blood and deep culture, surgical debridement, vacuum-assisted closure, and pathogen-targeted antibiotic treatment are the mainstays of treating surgical site infections. Surgical site infections are most caused by Gram-positive cocci such as *S. aureus* and *Enterococcus* spp., although Gram-negative pathogens such as *Escherichia coli*, *Acinetobacter baumannii*, and *Pseudomonas aeruginosa* can also cause surgical site infections [42,43,44,45]. Furthermore, polymicrobial site infections involving Gram-negative and/or Gram-positive pathogens and/or anaerobes are not uncommon [42,43,44,45].

### 3.3. Intra-Abdominal Infections

Early bacterial infections following liver transplantation are commonly caused by intra-abdominal infections, which account for 27–47% of all cases [4,31,35]. Cholangitis and peritonitis tend to be prevalent in the first few weeks following liver transplantation. Hepatic artery thrombosis, Roux-en-Y biliary reconstruction, and arterial stenosis all increase the risk of intra-abdominal infections [4,35]. Percutaneous or open surgical drainage, for the treatment of infected collections, is paired with prolonged antibiotic therapy, guided by susceptibility testing. *Enterococci*, *S. aureus*, and Gram-negative bacilli such as *Pseudomonas* sp., *Klebsiella* sp., *Acinetobacter* sp., and *Enterobacter* sp. are common causes of intra-abdominal infections [35].

### 3.4. Bloodstream Infections

Most bloodstream infections (BSI) arise in the first month following liver transplantation. Allograft rejection and intra-abdominal infection are two of the most common risk factors for this condition [4,31,35]. The gastrointestinal tract, urinary tract, lower respiratory tract, or infections from infected indwelling vascular catheters are the most prevalent sources of bloodstream infection after liver transplantation. Pathogen-directed antimicrobial therapy guided by antimicrobial susceptibility testing should be the goal of treating bloodstream infections, along with eradication of the predisposing factor [35]. Trans-esophageal echocardiography (TEE) should be used to diagnose endocarditis in patients with recurrent bloodstream infections. Removal of indwelling catheters, drainage of intraabdominal abscesses, and surgical correction of other nidi of infection are all necessary steps in the treatment of bacterial infections. On the one hand, *S.aureus*, *Enterococcus* spp., *Streptococcus viridans*, Gram-negative bacilli, or even a polymicrobial infection can cause bloodstream infection [35]. MRSA can cause up to 50% of BSI in liver transplant patients, which has serious implications for the efficacy of empirical therapy [46,47]. Prior to liver transplantation, *S. aureus* carriers have an increased risk of infection (24 to 87%) and may benefit from decontamination [46,47,48,49]. *S. aureus* infections may be reduced by identifying MRSA-infected individuals prior to donation and eradicating them [46,47,48,49]. A patient’s decolonization is not always permanent, so it is tough to decide the best time for it. Gram-negative bacteria, on the other hand, are an increasingly prevalent source of bloodstream infection, especially when they originate in the digestive system [39,40]. These Gram-negative pathogens are becoming more resistant. *E. coli* strains that have developed resistance to the quinolones ciprofloxacin and norfloxacin (which are commonly used as prophylaxis for spontaneous bacterial peritonitis, or as empiric therapy for community-acquired respiratory and urinary infections) and have a prevalence of nearly 13% in some centres [39,40,50,51,52,53]. Likewise, multi-drug resistant strains have been reported in as high as 62.5% of *A. baumannii*, 54.2% of *Stenotrophomonas maltophilia*, and 51.5% of *Pseudomonas* spp. isolates [39,40,50,51,52,53].

### 3.5. Mycobacterium tuberculosis and Non-Tuberculous Mycobacteria

*Mycobacterium tuberculosis* infection in LT has an annual impact of 450 cases in every 100,000 recipients [54]. The prevalence of *M. tuberculosis* infection varies widely depending on the geographical area (0.6% in the US, 1.4% in EU countries, 2.2% in non-US/non-EU countries) [1]. Pulmonary tuberculosis constitutes the majority of cases (about 60%), accordingly to a meta-analysis by Holty et al. [54]. The overall mortality rate was 18% and rose to 36% if *M. tuberculosis* infection was diagnosed in the first 5 months after LT [54]. Holty et al. reported that isoniazid prophylaxis (≥6 months) in patients with risk (positive tuberculin skin test, positive Quantiferon for *M. tuberculosis* test or compatible X-ray lung findings) for latent *M. tuberculosis* infection (LTBI) significantly reduced the risk of reactivation in LT [54,55]. Despite that, tighter liver enzyme monitoring is needed, and careful monitoring for adverse effects is highly recommended in patients with LTBI attending LT during isoniazid or rifampicin prophylaxis [54,55]. In addition, in managing transplant recipients with tuberculosis, the interaction between anti-tuberculous and immunosuppressive medicines, which may increase the risk of graft rejection, is a key concern [56,57]. Despite that, due to the high mortality risk of active or reactivated *M. tuberculosis* infection, a tuberculin skin or Quantiferon test for *M. tuberculosis* alongside chest X-ray screening should be considered in all those who are waiting for an LT to allow early prophylaxis in case of positive screening test [54,56,57,58].

Non-tuberculous mycobacteria (NTM) are ubiquitous microorganisms, and more than 140 species of this group are known [59]. Unfortunately, the epidemiology of NTM in LT is not fully defined, and clinical presentations vary widely, resulting in pleuropulmonary, skin and soft tissue, lymphadenitis, bloodstream or disseminated infections [59,60]. Despite the theoretical protean presentation in LT of NTM infections, most cases reported in the literature included pleural, lung or cutaneous involvement, and the *Mycobacterium avium* complex microorganisms are the most frequent NTM in the case series [59,60]. Currently, there is no consensus about the need for NTM prophylaxis, and an active approach to diagnosis, including histopathologic examination and acid-fast bacilli culture of aspirates or biopsy specimens from affected areas, is required [61].

## 4. Fungal Infection: *Pneumocystis jirovecii*

*Pneumocystis jirovecii* is a fungus that causes interstitial pneumonia in immunocompromised HIV-positive and HIV-negative patients: *P. jirovecii* pneumonia (PJP) may even cause life-threatening respiratory failure [62]. Regarding liver transplantation, the incidence of PJP is around 7% of all cases of pneumonia, with mortality ranging from 7% to 88% [63]. However, the overall incidence of PJ infection has fallen in recent years due to fewer immunosuppressive regimens used after LT, and according to recent data, the incidence in recipients that undergo prophylaxis is below 2% and <1% in the first year after LT. Contrarily, the incidence rises to 11% in patients without prophylaxis [63,64]. Considerations for PJP prophylaxis include the degree of immunosuppression preceding LT, the duration and form of post-LT immunosuppression, the absolute lymphocyte count, and the presence of a concurrent solid organ transplant. In addition, in this patient cohort, prophylaxis should be explored for patients undergoing therapy for acute rejection events [63,64]. Most cases of PJP occurred in the first 7 months after LT [63], although Fortea et al., in their survey, recently showed a not insignificant number of PJ infections occurring beyond the usually recommended period of prophylaxis [65]. Currently, there is no consensus and a wide heterogeneity between centres about the need for and duration of PJP prophylaxis [63,64].

## 5. Viral Infections

Viral infections are well recognized complications of immunosuppression and can occur from the donor (donor-derived infections), reactivation of endogenous latent virus, nosocomial sources or from the community. Certain viruses, particularly herpes viruses and polyomavirus, hinder host defences, thus raising the risk of other infections. Viral infections are also considered as factors for acute and chronic rejection responses [66].

Particular viruses, such as EBV and HHV-8, can cause post-transplant lymphoproliferation and/or cervical cancers (papillomavirus). Other viruses, such influenza, are mainly acquired by environmental transmission. All of these conditions are easily detectable in their early stages and can be effectively treated [66].

### 5.1. Hepatitis B Virus

The progress in anti-HBV therapy during the last two decades has significantly improved the management of hepatitis B virus (HBV) patients before and after LT. The development of antiviral resistance with virological breakthrough and hepatitis flare was a recognized challenge to successful treatment of chronic hepatitis B with first-generation nucleos(t)ide analogues (NA). However, the selection of third-generation NAs (entecavir, tenofovir), which are characterized by a high barrier to resistance, provides the best chance of achieving long-term treatment goals [67].

Patients with compensated or decompensated cirrhosis need treatment, with any detectable HBV DNA level and regardless of ALT levels, and HBV DNA negativization before LT reduces the risk of HBV recurrent disease post-transplant [67].

After LT, anti-HBV prophylaxis ought to be lifelong, with the aim of preventing graft reinfection (also from extrahepatic reservoirs) and graft loss; the choice of NA will depend on the drugs used before surgery, and on the presence of drug-resistance mutations [68,69].

The mainstay of long-term prophylaxis against HBV recurrence is a combination of oral NA with low doses of immunoglobulin against HBV (HBIg), through different routes of administration.

However, the high costs of HBIg have required new strategies in order to minimize prophylaxis, particularly in those patients who achieve undetectable HBV DNA before LT [70,71].

Many studies showed that low doses of HBIg (with a tailored approach based on individual risks) [72,73,74,75,76,77,78] in combination with NA can optimize the cost-effectiveness of the prophylaxis, but also, discontinuation of HBIg after a defined period while continuing NA could be a successful and cheap option. Fung et al. [79] demonstrated that long-term entecavir monotherapy is highly effective in preventing HBV reactivation after LT with a durable HBsAg seroclearance rate of 92%, an undetectable HBV DNA rate of 100% at 8 years, and excellent long-term survival of 85% at 9 years.

However, patients with active hepatocellular carcinoma (HCC) or decompensated cirrhosis may need an LT before the achievement of undetectable serum HBV DNA. In this special setting, a more conservative approach with a long-term, low dose of HBIg in combination with NA could be used. 

Active immunization using HBV vaccines post-transplantation is an attractive alternative to frequent HBIg injections or infusions. This strategy has yielded conflicting results and, therefore, remains controversial.

Future strategies will include targeting different sites of the HBV replication cycle and restoring the host immunity response to attain complete viral eradication.

### 5.2. Cytomegalovirus

The immunomodulatory effects of CMV, mediated by impaired T cell and phagocytic function and cytokine dysregulation, can lead to opportunistic infections, rejection, graft loss, and in some cases reduced survival [55,56]. Following primary infection, most immunocompetent individuals remain asymptomatic. No contraindications exist for organ donation in the case of a donor with latent CMV infection [80,81,82]. De novo infection by a graft in naïve recipients, as well as reactivation of a latent infection in the recipient, should be avoided by specific anti-viral prophylaxis or virological monitoring and pre-emptive therapy. Most CMV-active anti-viral agents are, at least partially, effective in preventing/treating other herpes viruses—including HSV and VZV—but not all, e.g., letermovir. Recipient morbidity increases in the case of donor-seropositive and recipient-seronegative (D+/R−) combinations [83].

### 5.3. Herpes Viruses

HSV 1 and 2, VZV, and/or HHV-6 infect most individuals globally, and these infections can result in considerable morbidity and mortality if immunosuppression is performed after SOT [59,60,84,85,86]. Loss of regulatory cellular immunity facilitates revival of latent herpesviruses, which can happen after primary infection [61,87]. No contraindication to organ donation exists for donors presenting with latent herpes-family viral infections [82]. Some transplant centres perform retrospective, additional donor tests for latent HSV or VZV in cases of seronegative recipients (mostly children) to decide on specific anti-viral prophylaxis or treatments and follow-up. However, no evidence exists to suggest this, based on a few case reports [62,63,64,65].

### 5.4. COVID-19

In liver transplant (LT) candidates, COVID-19 infection is associated with a high mortality rate (32.7%), reaching 45% in patients with decompensated cirrhosis, MELD score ≥ 15 and dyspnoea on presentation; respiratory failure was the major cause of death. No significant differences were observed between first and second waves of the pandemic [88].

In LT recipients, the most common presenting symptoms of COVID-19 were fever, cough, and shortness of breath [89,90]. Most of the hospitalized patients had radiological signs of viral pneumonia and required respiratory support (oxygen supplementation 59%, non-invasive ventilation 22%, and mechanical ventilation 15%). Mortality was more common in male recipients and observed only in patients aged 60 years or older [67,68]. Independent risk factors for death were age > 70, diabetes and renal disease, while tacrolimus administration showed a positive independent influence on survival rates [91]. For this reason, in hospitalized patients, complete immunosuppression withdrawal should be discouraged, but mycophenolate dose reduction or withdrawal could help in preventing severe COVID-19 [92].

Vaccination against SARS-CoV-2 for patients with chronic liver diseases and transplant recipients is recommended as a priority in cirrhosis, decomposition of liver, or hepatobiliary cancer cases with high risk of severe COVID-19 [93]. In our experience, patients awaiting LT who underwent anti-COVID-19 vaccination with mRNA vaccine (Comirnaty (Pfizer-BioNTech, New York, NY, USA) or Moderna (Cambridge, MA, USA)), experienced a high seroconversion rate: 94.4% 23 days after vaccination (median IgG value: 1980 binding antibody units/mL) and 92.0% 68 days after vaccination (IgG value of 1450 binding antibody units/mL), without serious adverse events related to vaccination [94].

The Italian Transplant Authority, from November 2020, allowed the use of liver grafts from deceased donors with active SARS-CoV-2 infection in informed LT candidates, only if severely ill and with active/resolved COVID-19. In donors’ liver biopsy at transplantation, SARS-CoV-2 RNA tested negative in 100%, suggesting a very low risk of transmission with LT [73,74]. Notably, none of the recipients testing negative with the molecular test at transplantation was found to be SARS-CoV-2 positive during follow-up [95,96].

## 6. Current Guidelines for Prevention, Diagnosis and Therapeutic Management

The 7th edition of the Guide to the Quality and Safety of Organs for Transplantation is published by the European Directorate for the Quality of Medicines & HealthCare of the Council of Europe (EDQM). The guide aims to deliver support for professionals engaged in transplantation and donation, improve the quality and reduce the risk of complications [97]. In addition, the European Association for the Study of the Liver (EASL) Clinical Practice Guidelines were published to guide physicians in the evaluation and management of LT candidates, covering various aspects [98].

The American Society of Transplantation’s Infectious Diseases Community of Practice recommended, regarding MDR Gram-negative infections, that source control be crucial element in managing MDR Gram-negative infections [99]. For prevention, decolonization of recipients with known ESBL-EB colonization is not recommended. SOT recipients colonized or infected with ESBL-EB should not be disqualified from transplantation. Recommendations for diagnosis include the application of EUCAST or CLSI cephalosporin cut-offs to detect CRE and nucleic acid for producing carbapenemase to initiate infection control measures and lead empiric therapy. Finally, for treatment, combinations with carbapenem were related to better outcomes; however, carbapenem combinations are not suggested if the MIC of meropenem is ≥4 µg/mL. First-generation β-lactam-β-lactamase inhibitors, such as piperacillin tazobactam, should not be prescribed as first choice for ESBL-EB infections [99].

## 7. Immunosuppression in LT

The optimal management of immunosuppression drives optimal health after LT, avoiding both graft and systemic complications [100], ensuring patient and graft survival. Usually, the number and dosage of drugs are reduced over time, having as a goal in one hand, the normality of liver function tests and the low risk of allograft rejection, and in the other hand, the reduction of declining renal function, recurrent infections, de novo malignancy, and other toxicities due to excessive immunosuppression.

Immunosuppression after LT is divided into the induction and maintenance phases: the first usually consists of intravenous corticosteroids immediately posttransplant and/or anti IL-2 receptor antibodies. The maintenance phase is based on the calcineurin inhibitors (CNI); antiproliferative agents, such as mofetil mycophenolate or azathioprine, and everolimus can be used to lower the toxicity of CNI therapy or for those at higher risk of rejection, usually in combination with lower-dose CNI therapy [101].

A fixed scheme of post-LT immunosuppression has now been replaced by an individualized scheme tailored to the patient [102,103]. In critically ill patients, such as pre-LT infected recipients, induction therapy with basiliximab is recommended to delay the introduction of CNI (3–5 days after LT). In patients who develop post-LT infections, immunosuppression should be adjusted to the patient’s clinical condition, and in some cases, temporarily suspended. Early immunosuppression minimization after LT is an option, feasible in selected recipients, while complete withdrawal is successful in only a small proportion [104].

## 8. Surgical Frontiers of Ex Situ Liver Perfusion

For years, the organs harvested from deceased donors underwent static cold storage before implantation. In recent years, machine perfusions were developed that allow the dynamic preservation of organs, especially those coming from extended criteria donors. These machine perfusions carry the perfusate into the organ in hypothermic or normothermic conditions.

In hypothermic conditions the bacterial growth was not described; during normothermic machine perfusion (NMP) instead, the temperature and blood perfusate, supplemented with medications and nutrients, provide an ideal culture medium for microorganisms. Bruinsma et al. [105] demonstrated that the sub(normothermic) condition without antibiotics sustained the bacterial growth (*S. epidermidis* and *Staphylococcus aureus*) in standard solutions for cold storage. For these reasons, the use of antibiotics during NMP of the liver is well established and has been a part of each group’s clinical protocol. In the development phase of the study of Eshmuminov et al. [106], microbial contamination occurred in 69% of porcine livers with a mean occurrence of growth on 4 ± 1.6 perfusion days. So, they switched the antimicrobial infusion in a continuous application, reduced the perfusate sampling and avoided taking liver biopsies; as a result, no more microbiological growth was observed perfusing the livers ex situ up to 1 week.

In the last year, a new clinical syndrome after NMP, distinct from classical post-reperfusion syndrome, was described by Hann et al. (1 case) [107], and Nasralla et al. (4 cases) [108]. This syndrome is characterized by a delayed vasoplegia, occurring 5 to 10 min after reperfusion, that can take a long time to resolve (from 1 to 36 h), and is related to the action of endotoxins. In all 5 cases, pathogenic organisms rapidly grew in the machine perfusate and in the recipient’s blood within 24 h; this condition was completely restored after an appropriate antibiotic therapy, and none of the recipients needed re-transplantation. 

The current goal is to consider NMP as a new strategy to protect extended criteria grafts from the deleterious effects of static cold storage and ischemia-reperfusion injury. Moreover, it represents a platform for therapeutic interventions [78,79,80,81,82] during ex situ liver preservation capable of delivering therapeutic agents to improve ischemia reperfusion injury and liver steatosis, and to promote immunomodulation and endothelial protection. Another possibility could be to deliver, during NMP, targeted antimicrobial drugs in grafts from infected donors, testing the perfusate for microbiological growth before transplantation [109,110,111,112]. Goldaracena et al. [113] reported the possibility of the use, during NMP, of Miravirsen (an antisense miRNA that inactivated miRNA-122) in the induction of resistance to re-infection by hepatitis C virus in pig livers in a transplant setting. Liang et al. [114] inoculated *Escherichia coli* or carbapenem-resistant *Klebsiella pneumoniae* in a rat model to investigate if antibiotics combined with static cold storage or hypothermic machine perfusion or NMP could eliminate the bacteria. The use of antibiotics significantly reduced the bacterial load in hypothermic as well as in NMP groups. When grafts infected with carbapenem-resistant *Klebsiella pneumoniae* were transplanted, the rats in the static cold storage group developed severe infection after transplantation, while those in the hypothermic machine perfusion group were in good condition.

## 9. Discussion

Infectious complications in liver transplant recipients are considered a primary risk factor for higher morbidity and mortality. Therefore, the patient at risk should be screened at baseline to define the risk of a subsequent infection. Screening starts from the past medical history, previous hospitalization, surgical interventions, and risk of immunosuppression other than liver failure-related immunodepression.

In addition, risk factors vary in the early period versus the late period after LT. Early risk factors include surgical practices and invasive procedures (i.e., mechanical ventilation, indwelling vascular and urinary catheterization). Late risk factors include mismatch status for viruses, allograft rejection, donor-transmitted diseases, repeated biliary tract manipulations and re-transplantation.

In the era of MDR pathogens, screening should be focused on microbiological colonization and the history of antimicrobial exposure. Following updated Italian recommendations, CRE colonization should not be viewed as an absolute contraindication for organ donation unless the colonization is in the organ to be transplanted, given the low risk of CRE transmission from the donor to the recipient [115].

The high risk of invasive infections in recipients colonized by MRSA [47,48] or KPC [7,8] commonly increases morbidity and mortality. In patients with CRE, BSI conferred a significantly lower survival rate [7,85,86]. Despite that, patients’ decolonization is not always recommended. New antibiotics that have been approved recently for CRE infections, such as ceftazidime-avibactam, meropenem-vaborbactam, imipenem-relebactam and cefiderocol, are greatly assisting in the management of those infections with favourable efficacy and safety profiles [87,88]. The timing of transplantation and burden of immunosuppression are other crucial factors when deciding to perform transplants in these patients.

After liver transplantation, the risk of bacterial infections fluctuates over time, and clinicians should be aware of the differences between early infections (<1 month) with a high risk of community-acquired infections and latent infections in which the risk of opportunistic infections and hospital-acquired pathogens increases with prolonged immunosuppression and hospitalization (Figure 1) [116,117,118,119,120].

Enhancing infection control practices continue to be a vital element for decreasing infections in transplantation [10,89]. General perceptions among healthcare workers about the possible adverse events of infections in this population and the significance of prompt detection of infection greatly influence the prevalence of infections and subsequent complications. Particularly, hand hygiene and contact isolation will assist in decreasing nosocomial infections. Devoted antimicrobial stewardship and specific bundles upon the local epidemiology and demands are recommended [11,12,13]. Antimicrobial stewardship programs among transplant patients have demonstrated encouraging effects, despite a lack of research on the topic [11,12,13]. Multidisciplinary teams of transplant surgeons, hepatologists, infectious disease specialists, microbiologists, and pharmacologists are essential to the success of these efforts.

## 10. Conclusions

Liver transplantation is a lifesaving treatment for acute and chronic end-stage liver diseases. Infections in transplant recipients account for the main cause of mortality and morbidity with rates of up to 80% [6,7,8] despite advanced surgical techniques, new immunosuppressive drugs, prophylactic antibiotics, vaccination, and infection control strategies. The increasing incidence of MDR infections has raised clinical needs in terms of infection control strategies and empiric and targeted antibiotic and antiviral therapies. Nowadays, the presence of a multidisciplinary team of transplant surgeons, hepatologists, infectious disease specialists, microbiologists, and pharmacologists is essential to the success of liver transplantation due to the progressively increasing complexities in the management of these patients.

## Figures and Tables

**Figure 1 biomedicines-10-01561-f001:**
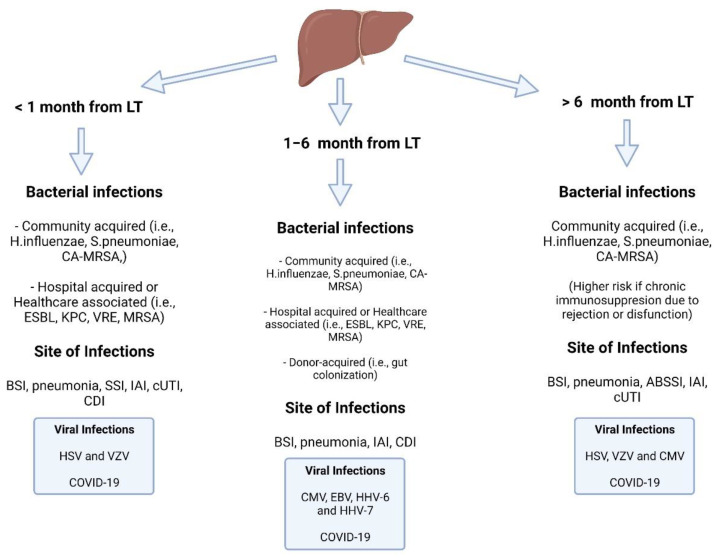
Timeline of bacterial and viral infections in the liver transplant recipient. Abbreviations: LT: liver transplant; CA-MRSA: community-acquired methicillin-resistant Staphylococcus aureus; ESBL: extended-spectrum beta-lactamase; KPC: Klebsiella pneumoniae carbapenemase; VRE: vancomycin-resistant Enterococcus; BSI: bloodstream infection; SSI: surgical site infection: IAI: intra-abdominal infection; cUTI: complicated urinary tract infection; CDI: *C. difficile* infection; HSV: herpes simplex virus: VZV: varicella zoster virus; EBV: Epstein-Barr virus; HHV: human herpes virus; ABSSI: acute bacterial skin and soft tissue infection.

**Table 1 biomedicines-10-01561-t001:** Risk assessment in solid organ transplant (SOT).

Pre-Transplant	Transplant	Post-Transplant
Epidemiological surveillance Outbreak investigation Screening programs Vaccination Assessment of medical history, sexual activity and drug use Water management plans and *Legionella* screening	Isolation and cohorting Personal protective equipment Hand hygiene Management of invasive devices Sterilization and disinfection High efficiency particulate air (HEPA) filter	Frequent air exchanges Airway duct policies Construction restrictions Environmental controls Multidisciplinary collaboration in follow-up management

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
