# Peer review of "Bacterial and Viral Infections in Liver Transplantation: New Insights from Clinical and Surgical Perspectives"

_biomedicines, 2022, doi:10.3390/biomedicines10071561_

Round 1
Reviewer 1 Report
In this manuscript, the authors have clarified the importance of infections after liver transplantation (LT). However, the extent of novelty seems to be lacking.
The prevalence of infection and infection-associated mortality after LT depends on the extent of immunosuppression and can reach 80% depending on the fear of rejection. However, this is iatrogenic.
The importance of the article, in my view, could benefit from discussion of the extent of immunosuppression and individual approach.
Viral infections such as HSV, VZV, HHV-6, 7 are overrepresented and almost never occur if immunosuppression is not overdone.
Author Response
Dear reviewer,
Thank you for your valuable comments.
We have added a paragraph about the immunosuppression in this setting.
We have indicated in the paragraph of HSV, VZV that they occur only if immunosuppression is done and we have summarized the paragraph in order to avoid over expression.
Reviewer 2 Report
This is a comprehensive review on the effects of bacterial and viral infection on liver transplantation. For viral infection, just wonder if hepatitis B is an important factor and shall be included. For clinical beneficial, it would be better to include current guidelines of prevention, diagnosis and therapeutic management from major countries on this aspect for clinician to refer.
Author Response
Dear reviewer,
Thank you for your valuable comments.
We have added a paragraph about HBV in this setting.
We have also added current guidelines from governing bodies in the field about prevention, diagnosis and management.
Reviewer 3 Report
it is an excellent review. Well organized, well written and absolutely comprehensive. It includes new information about Covid-19 infection in patients undergoing liver transplantation and a paragraph regarding new methods of liver graft storage before the transplantation.
I believe that 2 paragraphs are missing. The 1st one about pneumonocystis jirovevii infection (i believe that it must be included, though this pathogen is considered a fungus) and a second one about mycobacterium tuberculosis and mycobacterium avium complexes as well.
Author Response
Dear reviewer,
Thank you for your valuable comments.
We have added paragraphs about p.jirovevii and mycobacterium TB abd avium in this setting.